# Set-based Neural Network Encoding

## Abstract

We propose an approach to neural network weight encoding for generalization performance prediction that utilizes set-to-set and set-to-vector functions to efficiently encode neural network parameters. Our approach is capable of encoding neural networks in a modelzoo of mixed architecture and different parameter sizes as opposed to previous approaches that require custom encoding models for different architectures. Furthermore, our **S**et-based **N**eural network **E**ncoder (SNE) takes into consideration the hierarchical computational structure of neural networks by utilizing a layer-wise encoding scheme that culminates to encoding all layer-wise encodings to obtain the neural network encoding vector. Additionally, we introduce a *pad-chunk-encode* pipeline to efficiently encode neural network layers that is adjustable to computational and memory constraints. We also introduce two new tasks for neural network generalization performance prediction: cross-dataset and cross-architecture. In cross-dataset performance prediction, we evaluate how well performance predictors generalize across modelzoos trained on different datasets but of the same architecture. In cross-architecture performance prediction, we evaluate how well generalization performance predictors transfer to modelzoos of different architecture. Experimentally, we show that SNE outperforms the relevant baselines on the cross-dataset task and provide the first set of results on the cross-architecture task.

## 1 Introduction

Recently, deep learning methods have been applied to a wide range of fields and problems. With this broad range of applications, large amounts of datasets are continually being made available in the public domain together with neural networks trained on these datasets. Given this abundance of trained neural network models, the following curiosity arises: what can we deduce about these networks with access only to the parameter values? More generally, can we predict properties of these networks such as generalization performance on a testset, the dataset on which the model was trained, the choice of optimizer and learning rate, the number of training epochs, choice of model initialization etc. through an analysis of the model parameters? The ability to infer such fundamental properties of trained neural networks using only the parameter values has the potential to open up new application and research paradigms.

In this work, we tackle a specific version of this problem, namely, that of predicting the generalization performance on a testset of a neural network given access, only to the parameter values at the end of the training process. The first approach to solving this problem, proposed by Unterthiner et al. (2020), involves computing statistics such as the mean, standard deviation and quantiles, of each layer in the network, concatenating them to a single vector that represents the neural network encoding, and using this vector to predict the performance of the network. Another approach, also proposed as a baseline in Unterthiner et al. (2020), involves flattening all the parameter values of the network into a single vector which is then fed as input to layers of multilayer perceptrons(MLPs) to predict the network's performance. An immediate consequence of this approach is that it is practical only for moderately sized neural network architectures. Additionally, this approach ignores the hierarchical computational structure of neural networks through the weight vectorization process. The second, and most recent approach to this problem, proposed by Zhou et al. (2023), takes a geometric approach to the problem by building neural network weight encoding functions, termed neural functionals, that respect symmetric properties of permutation invariance and equivariance of the hidden layers of multilayer perceptrons under the action of an appropriately applied permutation group. While this approach respect these fundamental properties in the parameter space, it's application is restricted,

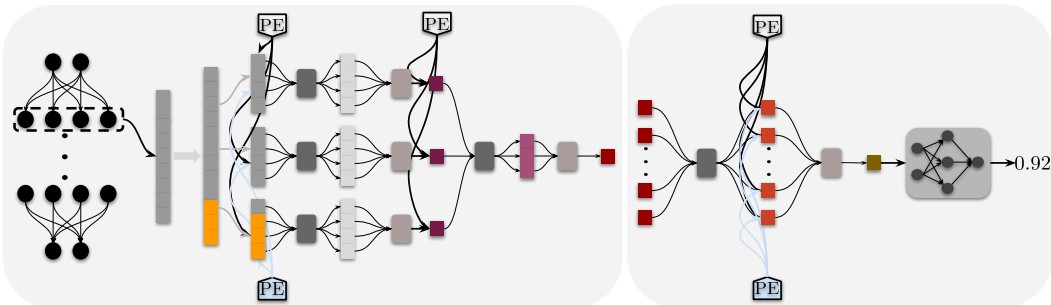

Figure 1: **Legend:** ■: Padding, ■: Set-to-Set Function, ■: Set-to-Vector Function, PE: Layer-Level Encoder, PE: Layer-Type Encoder. **Concept:** *(left)* Given the weights of a layer, SNE begins by padding and chunking the weights into *chunksizes*. Each chunk of the layer weight goes through a series of set-to-set and set-to-vector functions to obtain the chunk representation vector. Layer *level* and layer *type* positional encodings are used to inject structural information of the network at each stage of the chunk encoding process. All chunk encoding vectors are encoded together to obtain the layer encoding. *(right)* All layer encodings in the neural network are encoded to obtain the neural network encoding vector again using as series of set-to-set and set-to-vector functions. This vector is then used to predict the generalization performance of the neural network.

strictly, to multilayer perceptrons. Also, even when relaxations are made to extend this method to convolutional networks and combinations of convolutional layers and multilayer perceptrons, these only work under strict conditions of equivalence in the channel size in the last convolutional layer and the first linear layer. Hence it is clear that while the method proposed by Zhou et al. (2023) enjoys nice theoretical properties, its application is limited to only a small subset of carefully chosen architectures.

Moreover, both approaches (Unterthiner et al., 2020; Zhou et al., 2023) have a fundamental limitation: their encoding methods are applicable only to a single fixed, pre chosen neural network architecture. Once the performance predictor is trained, in the case of Unterthiner et al. (2020), and the neural network encoder of Zhou et al. (2023) is defined, they cannot be used to predict the performance of neural networks of different architecture. Consequently, evaluating these models on diverse architectures is infeasible without training an new generalization performance predictor for each architecture.

To this end, we propose a Set-based Neural Network Encoder (SNE) for predicting the performance of neural networks given only the model parameters that is agnostic to the network architecture. Specifically, we treat the neural network encoding problem from a set encoding perspective by utilising compositions of *set-to-set* and *set-to-vector* functions. However, the parameters of neural networks are ordered. To retain this order information, we utilize positional encoding Vaswani et al. (2017) at various stages in our model. Also, our model incorporates the hierarchical computational structure of neural networks in the encoder design by encoding independently, layer-wise, culminating in a final encoding stage that compresses all the layer-wise information into a single encoding vector used to predict the network performance. To handle the issue of large and variable parameter sizes efficiently, we incorporate a *pad-chunk-encode* pipeline that is parallelizable and can be used to iteratively encode layer parameters. In terms of evaluation, we introduce two new tasks: cross-dataset neural network performance prediction and cross-architecture neural network performance prediction. In cross-dataset neural network performance prediction, we fix the neural network architecture used to generate the training data and evaluate how well the performance predictors transfer to the same architecture trained on different datasets. For cross-architecture neural network performance prediction, we fix only the architecture for generating the training data and evaluate the performance of the predictors on architectures unseen during training.

Our contributions are as follows:

- We develop a Set-based Neural Network Encoder (SNE) for predicting the performance of neural networks given access only to parameter values that is capable of encoding neural networks of arbitrary architecture and takes into account the hierarchical computational structure of neural networks.

- We introduce the cross-dataset neural network performance prediction task where we evaluate how well neural network performance predictors transfer across neural networks trained on different datasets.

- We introduce the cross-architecture neural network performance prediction task where we evaluate how well neural network performance predictors trained on a specific architecture transfer to unseen architectures during training.

- We benchmark our method, SNE, against the relevant baselines on the cross-dataset task and show significant improvement over the baselines.

- Finally, we provide the first set of results on the cross-architecture task using our set-based neural network encoder, SNE.

## 2 RELATED WORK

**Set Functions:** Neural networks that operate on set structured data have recently been used in many applications ranging from point cloud classification to set generation (Kim et al., 2021). Set functions are required to respect symmetric properties such as permutation invariance and equivariance. In DeepSets (Zaheer et al., 2017), a set of sum-decomposable functions are introduced that are equivariant in the Set-to-Set applications and invariant in the Set-to-Vector applications. In Set Transformers (Lee et al., 2019), a class of attention based Set-to-Set and Set-to-Vector functions are introduced that are more expressive and capable of modeling pairwise and higher order interactions between set elements. Recent works such as Bruno et al. (2021) and Willette et al. (2023) deal with the problem of processing sets of large cardinality in the the limited memory/computational budget regime. In this work, we utilize the class of set functions developed in Lee et al. (2019) to develop a neural network encoder for performance prediction that is agnostic to specific architectural choices. Our set-based formulation allows us to build such an encoder, capable of handling neural networks weights of arbitrary parameter sizes. This is different from recent approaches to neural network encoding for performance prediction that can encode only parameters of a single architecture.

**Neural Network Performance Prediction From Weights:** Predicting the performance of neural networks given access only to the trained parameters is a relatively new topic of research introduced by Unterthiner et al. (2020). In Unterthiner et al. (2020), two methods are proposed for predicting the generalization performance of neural networks: the first involves flattening the weights of the network into a single vector, processing it using multiple layers of MLPs to obtain an encoding vector which is then used to predict the performance. The second involves computing the statistics of each layer in the network, such as mean, variance, quantiles etc., concatenating them into a single vector that is then used for predicting the performance of the network. The most recent approach that we are aware of, Navon et al. (2023) and Zhou et al. (2023), proposes a neural network weight encoder that is invariant or equivariant, depending on the application, to an appropriately applied permutation group to the hidden layers of MLPs. Two variants of their model is provided: one which operates only on the hidden layers, and conforms strictly to the theory of permuting MLP hidden neurons (Hecht-Nielsen, 1990), and a relaxation that assumes that the neurons of both the input and output layers of MLPs are permutable. Additionally, extensions are provided for convolutional layers. Our approach, SNE, is directly comparable to these methods for the neural network performance prediction task. However, unlike the methods of Unterthiner et al. (2020), Navon et al. (2023) and Zhou et al. (2023) which operate only on neural networks of fixed architecture, and consequently fixed number of parameters, SNE is capable of encoding networks of arbitrary architecture. Moreover, SNE utilizes the hierarchical computation structure of neural networks by encoding, iteratively or in parallel, from the input to the output layers. Furthermore, we go further than the experimental evaluation in Unterthiner et al. (2020) and Zhou et al. (2023) by introducing two new tasks: cross-dataset and cross-architecture neural network performance prediction. Unterthiner et al. (2020) and Zhou et al. (2023) can only be benchmarked on the cross-dataset task where all networks in the modelzoos are of the same architecture. Their restriction to a single fixed architecture makes cross-architecture evaluation impossible. Our method, SNE, on the other hand can be used for both tasks.

## 3 SET-BASED NEURAL NETWORK ENCODING

### 3.1 PRELIMINARIES

We have access to a dataset $D = \{(x_1, y_2), \ldots, (x_n, y_n)\}$ where for each $(x_i, y_i)$ pair, $x_i$ represents the weights of a neural network architecture $a$, sampled from a set of architectures $\mathcal{A}$ and $y_i$

corresponds to some property of $x_i$ after it has been trained on a specific dataset $d$. $y_i$ can be properties such as generalization gap, training loss, the learning rate used to train $x_i$, or even the number of epochs, choice of weight initialization, and optimizer used to train $x_i$. Henceforth, we refer to $D$ as a *modelzoo*. For each $x_i \in D$, $x_i = [w_i^0, \ldots, w_i^{|x_i|}]$ where $w_i^j$ represents the weights (parameters) of the $jth$ layer of the neural network $x_i$, and $|x_i|$ is the total number of layers in $x_i$. Consequently, $w_i^0$ and $w_i^{|x_i|}$ are the input and output layers of $x_i$ respectively. Additionally, we introduce the $\texttt{Flatten} : x_i \to \mathbf{R}^{d_i}$ operation, that takes as input the weights of a neural network and returns the flattened weights and $d_i$ is the total number of parameter is $x_i$.

The neural network encoding problem is defined such that, we seek to compress $x_i \in \mathbf{R}^{d_i}$ to a compact representation $z_{x_i} \in \mathbf{R}^h$ such that $z_{x_i}$ can be used to predict the properties $y_i$ of $x_i$ with $h \ll d_i$. In what follows, we present the details of our Set-Based Neural Network Encoding (SNE) method capable of encoding the weights of neural networks of arbitrary architecture that takes into account the hierarchical computational structure of the given architecture and with efficient methods for processing weights of high dimension.

## 3.2 Handling High Dimensional Layer Weights via Chunking

For a given layer $w_i^j \in x_i$, the dimension of $w_i^j$, $|w_i^j|$ can be very large. For instance, when considering linear layers, flattening the weights can results in a tensor that can require large compute memory to be processable by another neural network. To resolve this issue, we resort to *chunking*. Specifically, for all layers $w_i^j \in x_i$, we perform the following operations:

$$\hat{w}_i^j = \texttt{Chunk}(\texttt{Pad}(\texttt{Flatten}(w_i^j), c), c) = \{w_i^{j_0}, \ldots, w_i^{j_q}\}, \tag{1}$$

where for any $w_i^{j_t} \in \hat{w}_i^j$, $|w_i^{j_t}| \in \mathbf{R}^c$. Here, $c$ is the *chunksize*, fixed for all layer types in the neural network and $t \in [0, \ldots, q]$. The padding operation $\texttt{Pad}(w_i^j, c)$ appends zeros, if required, to extend $w_i^j$ and make its dimension a multiple of the chunksize $c$. To distinguish padded values from actual weight values, each element of $\hat{w}_i^j$ has a corresponding set of masks $\hat{m}_i^j = \{m_i^{j_0}, \ldots, m_i^{j_q}\}$. Note that with this padding and subsequent chunking operation, each element in $\hat{w}_i^j$ is now small enough, for an appropriately chosen chunksize $c$, to be processed. Moreover, all the elements in $\hat{w}_i^j$ can be processed in parallel.

The modelzoos we consider in the experimental section are populated by neural networks with stacks of convolutional and linear layers. For each such layer, we apply the padding and chunking operation differently. For a linear layer $w_i^j \in \mathbf{R}^{\texttt{out} \times \texttt{in}}$, where $\texttt{out}$ and $\texttt{in}$ are the input and output dimensions respectively, we apply the flattening operation on both dimensions followed by padding and chunking. However for a convolutional layer $w_i^j \in \mathbf{R}^{\texttt{out} \times \texttt{in} \times \texttt{k} \times \texttt{k}}$, we apply the flattening, padding, and chunking operations only to the kernel dimensions $\texttt{k}$.

Finally we note that for layers with bias values, we apply the procedure detailed above independently to both the weights and biases.

## 3.3 Independent Chunk Encoding

The next stage in our Set-based Neural Network encoding pipeline involves encoding, independently, each chunk of weight in $\hat{w}_i^j = \{w_i^{j_0}, \ldots, w_i^{j_t}\}$. For each $w_i^{j_t} \in \hat{w}_i^j$, we treat the $c$ elements as members of a set. However, it is clear that $w_i^{j_t}$ has order in its sequence, *i.e.*, an ordered set. We remedy this by providing this order information via positional encoding. Concretely, for a given $w_i^{j_t} \in \mathbf{R}^{c \times 1}$, we first model the pairwise relations between all $c$ elements using a *set-to-set* function $\Phi_{\theta_1}$ to obtain:

$$\hat{w}_i^{j_t} = \Phi_{\theta_1}(w_i^{j_t}) \in \mathbf{R}^{c \times h}. \tag{2}$$

That is, $\Phi_{\theta_1}$ captures pair-wise correlations in $w_i^{j_t}$ and projects all elements (weight values) to a new dimension $h$.

Given $\hat{w}_i^{j_t} \in \mathcal{R}^{c \times h}$, we inject two kinds of positionally encoded information. The first encodes the *layer type* in a list of layers, *i.e.*, linear or convolution for the modelzoos we experiment with, to obtain:

$$\hat{w}_i^{j_t} = \texttt{PosEnc}_{Layer}^{Type}(\hat{w}_i^{j_t}) \in \mathbf{R}^{c \times h}. \tag{3}$$

Here we abuse notation and assign the output of PosEnc($\cdot$) to $\hat{w}_i^{i_t}$ to convey the fact that $\hat{w}_i^{j_t}$'s are modified in place and to simplify the notation. Also, all PosEnc($\cdot$)s are variants of the positional encoding method introduced in Vaswani et al. (2017). Next we inject the layer level information. Since neural networks are computationally hierarchical, starting from the input to the output layer, we include this information to distinguish chunks, $w_i^{j_t}$s from different layers. Specifically, we compute:

$$\hat{w}_i^{j_t} = \text{PosEnc}_{Layer}^{Level}(\hat{w}_i^{j_t}) \in \mathbf{R}^{c \times h}, \tag{4}$$

where the input to $\text{PosEnc}_{Layer}^{Level}(\cdot)$ is the output of Equation 3. We note that this approach is different from previous neural network encoding methods (Unterthiner et al., 2020) that loose the layer/type information by directly encoding the entire flattened weights hence disregarding the hierarchical computational structure of neural networks. Experimentally, we find that injecting such positionally encoded information improves the models performance.

We further model pairwise correlations in $\hat{w}_i^{j_t}$, now infused with layer/type information, using another set-to-set function $\Phi_{\theta_2}$:

$$\hat{w}_i^{j_t} = \Phi_{\theta_2}(w_i^{j_t}) \in \mathbf{R}^{c \times h}. \tag{5}$$

The final step in the chunk encoding pipeline involves compressing all $c$ elements in $\hat{w}_i^{j_t}$ to a compact representation. For this, we use a *set-to-vector* function $\Gamma_{\theta_\alpha} : \mathbf{R}^{c \times h} \to \mathbf{R}^h$. In summary, the chunk encoding layer computes the following function:

$$\tilde{w}_i^{j_t} = \Gamma_{\theta_\alpha}[\Phi_{\theta_2}(\text{PosEnc}_{Layer}^{Level}(\text{PosEnc}_{Layer}^{Type}(\Phi_{\theta_1}(w_i^{j_t}))))] \in \mathbf{R}^{1 \times H}. \tag{6}$$

Note now that for each chunked layer $\hat{w}_i^j = \{w_i^{j_0}, \ldots, w_i^{j_q}\}$, the chunk encoder, Equation 6, produces a new set $\tilde{w}_i^j = \texttt{Concatenate}[\{\tilde{w}_i^{j_0}, \ldots, \tilde{w}_i^{j_q}\}] \in \mathbf{R}^{q \times h}$, which represents all the encodings of all chunks in a layer.

**Remark** Our usage of set functions $\Phi_{\theta_1}$, $\Phi_{\theta_2}$ and $\Gamma_{\theta_\alpha}$ allows us to process layers of arbitrary sizes. This in turn allows us to process neural networks of arbitrary architecture using a single model, a property lacking in previous approaches to neural network encoding for generalization performance prediction (Zhou et al., 2023; Unterthiner et al., 2020).

### 3.4 Layer Encoding

At this point, we have encoded all the chunked parameters of a given layer to obtain $\tilde{w}_i^j$. Encoding a layer, $w_i^j$, then involves defining a function $\Gamma_{\theta_\beta} : \mathbf{R}^{q \times h} \to \mathbf{R}^{1 \times h}$ for arbitrary $q$. In practice, this is done by computing:

$$\mathbf{w}_i^j = \Gamma_{\theta_\beta}[\text{PosEnc}_{Layer}^{Level}(\Phi_{\theta_3}(\tilde{w}_i^j))] \in \mathbf{R}^{1 \times h}. \tag{7}$$

Again we have injected the layer level information, via positional encoding, into the encoding processed by the set-to-set function $\Phi_{\theta_3}$. We then collect all the layer level encoding of the neural network $x_i$:

$$\tilde{w}_i = \texttt{Concatenate}[\mathbf{w}_i^0, \ldots, \mathbf{w}_i^{|x_i|}] \in \mathbf{R}^{|x_i| \times h}. \tag{8}$$

### 3.5 Neural Network Encoding

With all layers in $x_i$ encoded, we compute the neural network encoding vector $z_{x_i}$ as follows:

$$z_{x_i} = \Gamma_{\theta_\gamma}[\Phi_{\theta_4}(\text{PosEnc}_{Layer}^{Level}(\tilde{w}_i))] \in \mathbf{R}^h. \tag{9}$$

$z_{x_i}$ compresses all the layer-wise information into a compact representation for the downstream task. Since $\Gamma_{\theta_\gamma}$ is agnostic to the number of layers $|x_i|$ of network $x_i$, the encoding mechanism can handle networks of arbitrary layers and by extension architecture. Similar to the layer encoding pipeline, we again re-inject the layer-level information through positional encoding before compressing with $\Gamma_{\theta_\gamma}$.

Henceforth, we refer to the entire neural network encoding pipeline detailed so far as $\text{SNE}_\Theta(x_i)$ for a network $x_i$, where $\Theta$ encapsulates all the model parameters, $\Phi_{\theta_{1-4}}, \Gamma_\alpha, \Gamma_\beta$ and $\Gamma_\gamma$.

### 3.6 Choice of Set-to-Set and Set-to-Vector Functions

Now, we specify the choice of Set-to-Set and Set-to-Vector functions encapsulated by $\Phi_{\theta_{1-4}}, \Gamma_\alpha, \Gamma_\beta$ and $\Gamma_\gamma$ that are used to implement SNE. Let $X \in \mathbf{R}^{n_X \times d}$ and $Y \in \mathbf{R}^{n_Y \times d}$ be arbitrary sets where $n_X = |X|$, $n_Y = |Y|$ and $d$ (note the abuse of notation from Section 3.1 where $d$ is a dataset) is the dimension of an element in both $X$ and $Y$.

The MultiHead Attention Block (MAB) with parameter $\omega$ is given by:

$$\text{MAB}(X, Y; \omega) = \text{LayerNorm}(H + \text{rFF}(H)), \quad \text{where} \tag{10}$$

$$H = \text{LayerNorm}(X + \text{MultiHead}(X, Y, Y; \omega)). \tag{11}$$

Here, LayerNorm and rFF are Layer Normalization (Ba et al., 2016) and row-wise feedforward layers respectively. MultiHead$(X, Y, Y; \omega)$ is the multihead attention layer of Vaswani et al. (2017).

The Set Attention Block (Lee et al., 2019), SAB, is given by:

$$\text{SAB}(X) := \text{MAB}(X, X). \tag{12}$$

That is, SAB computes attention between set elements and models pairwise interactions and hence is a Set-to-Set function. Finally, the Pooling MultiHead Attention Block (Lee et al., 2019), PMA$_k$, is given by:

$$\text{PMA}_k(X) = \text{MAB}(S, \text{rFF}(X)), \quad \text{where} \tag{13}$$

$S \in \mathbf{R}^{k \times d}$ and $X \in \mathbf{R}^{n_X \times d}$. The $k$ elements of $S$ are termed *seed vectors* and when $k = 1$, as is in all our experiments, PMA$_k$ pools a set of size $n_X$ to a single vector making it a Set-to-Vector function.

All parameters encapsulated by $\Phi_{\theta_{1-4}}$ are implemented as a stack of two SAB modules: SAB(SAB($X$)). Stacking SAB modules enables us not only to model pairwise interactions but also higher order interactions between set elements. Finally, all of $\Gamma_\alpha, \Gamma_\beta$ and $\Gamma_\gamma$ are implemented as a single PMA module with $k = 1$.

### 3.7 Downstream Task

Given $(z_{x_i}, y_i)$, we train a predictor $f_\theta(z_{x_i})$ to estimate properties of the network $x_i$. In this work, we focus solely on the task of predicting the generalization performance of $x_i$, where $y_i$ is the performance on the test set of the dataset used to train $x_i$. The parameters of the predictor $f_\theta$ and all the parameters in the neural network encoding pipeline, $\Theta$, are jointly optimized. In particular, we minimize the error between $f_\theta(z_{x_i})$ and $y_i$. For the entire modelzoo, the objective is given as:

$$\underset{\Theta, \theta}{\text{minimize}} \sum_{i=1}^{d} \ell[f_\theta(\text{SNE}_\Theta(x_i)), y_i], \tag{14}$$

for an appropriately chosen loss function $\ell(\cdot)$. In our experiments, $\ell(\cdot)$ is the binary cross entropy loss. The entire SNE pipeline is shown in Figure 1.

## 4 Experiments

We now present experimental results on the cross-dataest and cross-architecture neural network performance prediction tasks. Details of experimental settings, hyperparameters, model specification etc. can be found in the Appendix.

### 4.1 Cross-Dataset Neural Network Performance Prediction

For this task, we train neural network performance predictors on 4 homogeneous modelzoos, of the same architecture, with each modelzoo specialized to a single dataset.

**Datasets and Neural Network Architecture:** Each modelzoo is trained on one of the following datasets: MNIST (Deng, 2012), FashionMNIST (Xiao et al., 2017), CIFAR10 (Krizhevsky, 2009) and SVHN (Netzer et al., 2018). We use the modelzoos provided by Unterthiner et al. (2020). To create each modelzoo, 30K different hyperparameter configurations were sampled. The hyperparameters include the learning rate, regularization coefficient, dropout rate, the variance and choice of initialization, activation functions etc. A thorough description of the modelzoo generation process

Table 1: Cross-Dataset Neural Network Performance Prediction. We benchmark how well each method transfers across multiple datasets. In the first column, $A \rightarrow B$ implies that a model trained on a homogeneous modelzoo of dataset $A$ is evaluated on a homogeneous modelzoo of dataset $B$. In the last row, we report the averaged performance of all methods across the cross-dataset task. For each row, the best model is shown in red and the second best in blue. Models are evaluated in terms of *Kendall's $\tau$* coefficient.

| | MLP | STATNN | NFN$_{NP}$ | NFN$_{HNP}$ | SNE(ours) |
|---|---|---|---|---|---|
| MNIST→ MNIST | 0.878±0.001 | 0.926±0.000 | 0.937±0.000 | **0.942±0.001** | 0.941±0.000 |
| MNIST→ FashionMNIST | 0.486±0.019 | 0.756±0.006 | 0.726±0.005 | 0.690±0.008 | **0.773±0.009** |
| MNIST→ CIFAR10 | 0.562±0.024 | 0.773±0.005 | 0.756±0.010 | 0.758±0.000 | **0.792±0.008** |
| MNIST→ SVHN | 0.544±0.005 | 0.698±0.005 | 0.702±0.005 | 0.710±0.010 | **0.721±0.001** |
| FashionMNIST→ FashionMNIST | 0.874±0.001 | 0.915±0.000 | 0.922±0.001 | **0.935±0.000** | 0.928±0.001 |
| FashionMNIST→ MNIST | 0.507±0.007 | 0.667±0.010 | **0.755±0.018** | 0.617±0.012 | 0.722±0.005 |
| FashionMNIST→ CIFAR10 | 0.515±0.007 | 0.698±0.029 | 0.733±0.007 | 0.695±0.032 | **0.745±0.008** |
| FashionMNIST→ SVHN | 0.554±0.006 | 0.502±0.043 | 0.663±0.014 | 0.662±0.003 | **0.664±0.003** |
| CIFAR10→ CIFAR10 | 0.880±0.000 | 0.912±0.001 | 0.924±0.002 | **0.931±0.000** | 0.927±0.000 |
| CIFAR10→ MNIST | 0.552±0.003 | 0.656±0.005 | **0.674±0.018** | 0.600±0.025 | 0.648±0.006 |
| CIFAR10→ FashionMNIST | 0.514±0.005 | **0.677±0.004** | 0.629±0.031 | 0.526±0.038 | 0.643±0.006 |
| CIFAR10→ SVHN | 0.578±0.005 | 0.728±0.004 | 0.697±0.006 | 0.662±0.004 | **0.753±0.007** |
| SVHN→ SVHN | 0.809±0.003 | 0.844±0.000 | 0.855±0.001 | **0.862±0.002** | 0.858±0.003 |
| SVHN→ MNIST | 0.545±0.025 | 0.630±0.009 | **0.674±0.008** | 0.647±0.016 | 0.647±0.001 |
| SVHN→ FashionMNIST | 0.523±0.026 | 0.616±0.007 | 0.567±0.014 | 0.494±0.023 | **0.655±0.003** |
| SVHN→ CIFAR10 | 0.540±0.027 | 0.746±0.002 | 0.725±0.007 | 0.547±0.039 | **0.760±0.006** |
| Average | 0.616±0.143 | 0.734±0.115 | 0.746±0.106 | 0.705±0.140 | **0.761±0.101** |

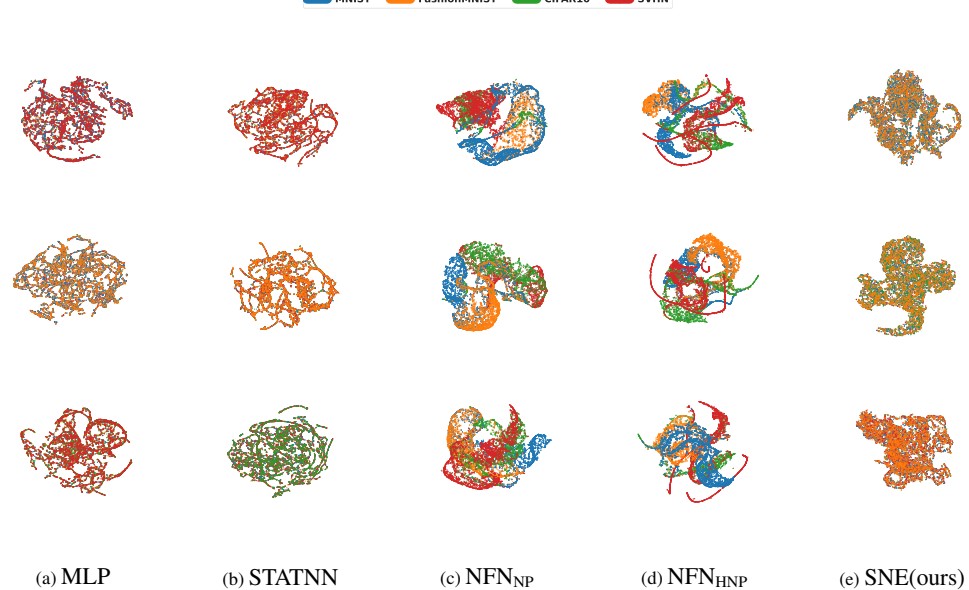

(a) MLP   (b) STATNN   (c) NFN$_{NP}$   (d) NFN$_{HNP}$   (e) SNE(ours)

Figure 2: TSNE Visualization of Neural Network Encoding. We train neural network performance prediction methods on a combination of the MNIST, FashionMNIST, CIFAR10 and SVHN modelzoos of Unterthiner et al. (2020). We present 3 views of the resulting 3-D plots showing how neural networks from each modelzoo are embedded/encoded by the corresponding models. Zoom in for better viewing.

can be found in Appendix A.2 of Unterthiner et al. (2020). The single architecture used to general the modelzoos consists of 3 convolutional layers each with 16 filters, a global average pooling layer and linear classification layer. Each modelzoo is split into a training, testing and validation splits.

**Task:** In this task, we consider cross-dataset neural network performance prediction where we evaluate the prediction performance on the testset of the modelzoo on which the predictors were trained on. Additionally, we evaluate how well each predictor transfers to the other modelzoos. To the

best of our knowledge, this is the first such empirical analysis of how well neural network performance predictors transfer to different datasets. We evaluate all models using *Kendall's* $\tau$ (Kendall, 1938), a rank correlation measure.

**Baselines:** We compare SNE with the following baselines:

- **MLP:** This model flattens all the weights and biases of each neural network into a single vector which is then fed to a stack of MLPs with ReLU activation and finally a Sigmoid activation function for predicting the performance of the neural network.

- **STATNN** (Unterthiner et al., 2020)**:** Given neural network weights, this model computes statistics of each layer, including their means, variance and quantiles, concatenates them into a single vector which is then used as input to a stack of MLPs with ReLU activation and a final layer with Sigmoid activation that outputs the performance of the neural network.

- **NFN$_{HNP}$ and NFN$_{NP}$** (Zhou et al., 2023)**:** These two models termed Neural Functionals(NF), developed mainly for MLPs, are designed to be permutation invariant or equivariant to an appropriately applied permutation group to the hidden layers of the neural networks. HNP, hidden neural permutation, is applied only to the hidden layers of each network since the output and input layers of MLPs are not invariant/equivariant to the action of a permutation group on the neurons. NP, neural permutation, makes a strong assumption that both the input and output layers are also invariant/equivariant under the action of a permutation group.

**Results:** We present the results of the cross-dataset neural network performance prediction task in Table 1. For each row in Table 1, the first column shows the cross-dataset evalutation direction. For instance, MNIST→CIFAR10 implies that a model trained on a modelzoo of neural networks trained on MNIST is cross evaluated on a modelzoo populated by neural networks trained on CIFAR10. We note that the A→A setting, *e.g.* MNIST→MNIST, corresponds to the evaluation settings of Unterthiner et al. (2020) and Zhou et al. (2023). Also in Table 1 we show the best model in red and the second best model in blue.

As show in Table 1, SNE is always either the best model or the second best model in the cross-dataset task. Especially, SNE is particularly good in the A→B performance prediction task compared to the next competitive baselines, NFN$_{NP}$ and NFN$_{HNP}$. The MLP baseline, as expected, performs the worse since concatenating all weight values in a single vector looses information such as the network structure. STATNN (Unterthiner et al., 2020) performs relatively better than the MLP baseline suggesting that the statistics of each layer indeed captures enough information to do moderately well on the neural network performance prediction task. NFN$_{NP}$ and NFN$_{HNP}$ perform much better than STATNN and MLP and NFN$_{HNP}$ in particular shows good results in the A→A setting. Interestingly, NFN$_{NP}$ is a much better cross-dataset performance prediction model than NFN$_{HNP}$. However, across the entire cross-dataset neural network performance prediction task, SNE significantly outperforms all the baselines as shown in the last row of Table 1.

Also, as stated in Section 3.3, positional encoding of layer and level information provides performance boost. For instance, removing all such encoding from SNE reduces the performance on CIFAR10→CIFAR10 from 0.928 to 0.918.

**Qualitative Analysis:** To further understand how SNE transfers well across modelzoos, we generate TSNE (Van der Maaten and Hinton, 2008) plots for the neural network encoding of all the compared models for all four homogeneous modelzoos in Figure 2. We provide 3 different views of each models embeddings to better illustrate the encoding pattern. In Figures 2c and 2d, we observe that NFN$_{NP}$ and NFN$_{HNP}$ have very clear separation boundaries between the networks from each modelzoo. This may explain NFN$_{HNP}$'s good performance in the A→A cross-dataset neural network encoding task in Table 1. In Figures 2a and 2b, MLP and STATNN, respectively show similar patterns with small continuous string of modelzoo specific groupings. However, these separations are not as defined as those of NFN$_{NP}$ and NFN$_{HNP}$. The embedding patter of SNE on the other hand is completely different. In Figure 2e, all networks from all the modelzoos are embedded almost uniformly close to each other. This may suggest why SNE performs much better on the cross-dataset performance prediction task since it is much easier to interpolate between the neural network encodings generated by SNE across modelzoos.

## 4.2 CROSS-ARCHITECTURE NEURAL NETWORK PERFORMANCE PREDICTION

For this tasks, we train SNE on 3 homogeneous modelzoos of the same architecture and test it on 3 other homogeneous modelzoos of a different architecture than the modelzoo used for training. The cross-architecture task demonstrates SNE's agnosticism to particular architectural choices since training and testing are done on modelzoos of different architectures.

**Datasets and Neural Network Architectures:** For cross architecture evaluation, we utilize 3 datasets: MNIST, CIFAR10 and SVHN. Since the modelzoos of Unterthiner et al. (2020) consist of a single architecture, we refer to them as $Arch_1$. We use all modelzoos of $Arch_1$ only for training the neural network performance predictors. We

Table 2: Cross-Architecture Neural Network Performance Prediction. We compare how SNE transfers across architectures.

| $Arch_1 \rightarrow Arch_2$ | SNE |
|---|---|
| MNIST→ MNIST | 0.452±0.021 |
| MNIST→ CIFAR10 | 0.478±0.010 |
| MNIST→ SVHN | 0.582±0.016 |
| CIFAR10→ CIFAR10 | 0.511±0.020 |
| CIFAR10→ MNIST | 0.467±0.020 |
| CIFAR10→ SVHN | 0.594±0.029 |
| SVHN→ SVHN | 0.621±0.013 |
| SVHN→ MNIST | 0.418±0.096 |
| SVHN→ CIFAR10 | 0.481±0.055 |

create another set of MNIST, CIFAR10 and SVHN modelzoos using a different architecture $Arch_2$. $Arch_2$ consists of 3 convolutional layers followed by two linear layers. Exact architectural specifications are detailed in the Appendix. We generate the modelzoos of $Arch_2$ following the routine described in Appendix A.2 of Unterthiner et al. (2020). All modelzoos of $Arch_2$ are used only for testing the cross-architecture neural network performance prediction of predictors trained on modelzoos of $Arch_1$ and are *not* used during training.

**Task:** For this task, we seek to explore the following question: Does neural network performance predictors trained on modelzoos of $Arch_1$ transfer or generalize on modelzoos of $Arch_2$? Also, we perform cross dataset evaluation on this task. However, this is different from the cross dataset evaluation in Section 4.1 since the cross evaluation is with respect to modelzoos of $Arch_2$.

**Baselines:** As we already remarked, none of the baselines used in the cross dataset task of Section 4.1 are applicable to this task. The MLP, STATNN, $NFN_{NP}$ and $NFN_{HNP}$ baselines all depend on the architecture of the modelzoo used for training and require modelzoos to be homogeneous architecturally. SNE on the other hand is agnostic to architectural choices and hence can be evaluated cross-architecture. Consequently, we provide the first set of results, to the best of our knowledge, on the cross-architecture neural network performance prediction task.

**Results:** We report the quantitative evaluation on the cross-architecture task in Table 2. The first column, $Arch_1 \rightarrow Arch_2$ shows the direction of transfer, where we train using modelzoos of $Arch_1$ and test on modelzoos of $Arch_2$. Additionally, A→B, *e.g.* MNIST→CIFAR10 shows the cross-dataset transfer as in Section 4.1. However, the transfer is now across architectures. We evaluate SNE in terms of Kendall's $\tau$.

From Table 2, it can be seen that SNE transfers well across architectures. Interestingly, the SVHN modelzoo, the most challenging modelzoo in the cross-dataset task, shows very good transfer across architectures. Alluding to the qualitative analysis in Section 4.1, we infer that SNE transfers well across architectures due to it's spread out neural network encoding pattern that allows for much easier interpolation even across unseen architectures as shown in Table 2.

## 5 CONCLUSION

In this work, we tackled the problem of predicting neural network generalization performance given access only to the trained parameter values. We presented a Set-based Neural Network Encoder (SNE) that reformulates the neural network encoding problem as a set encoding problem. Using a sequence of set-to-set and set-to-vector functions, SNE utilizes a pad-chunk-encode pipeline to encode each network layer independently; a sequence of operations that is parallelizable across chunked layer parameter values. SNE also utilizes the computational structure of neural networks by injecting positionally encoder layer type/level information in the encoding pipeline. As a result, SNE is capable of encoding neural networks of different architectures as opposed to previous methods that only work on a fixed architecture. Experimentally, we introduced the cross-dataset and cross-architecture neural network generalization performance prediction tasks. We demonstrated SNE's ability to transfer well across modelzoos of the same architecture but with networks trained on different datasets on the cross-dataset task. On the cross-architecture task, we demonstrated SNE's agnosticism to architectural choices and provided the first set of experimental results for this task.

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

## A    Organization

In Section B we provide some limitations and future directions for the neural network encoding task. In Section C, we specify all the model architectures used for generating the modelzoos of $Arch_1$ and $Arch_2$ used for the cross-dataset and cross-architecture tasks. In Section D we provide details of the train/validation/test splits. In Section E, we detail all the hyperparameters used for all experiments. Finally in we provide SNE implementation details is Section F.

## B    Limitations & Future Work

In this work, we focused solely on the task of predicting the properties, specifically generalization performance, of neural networks given access only to the model parameters. While the task of encoding neural network weights is a relatively new topic of research with very few baselines, we anticipate new applications/research directions where the neural network encoding vector is used for tasks such as neural network generation or neural network retrieval. These tasks and potential applications are out of the scope of this paper and we leave it for future works.

## C    Architectures for Generating Modelzoos

We specify the architectures for generating the modelzoos of $Arch_1$ and $Arch_2$. For the modelzoos of $Arch_1$ in Table 3, all datasets with with 3 channel images (CIFAR10 and SVHN) are converted to grayscale. This is in accordance with the setup of both Unterthiner et al. (2020) and Zhou et al. (2023) and allows us to evaluate both methods in the cross-dataset task for this set of homogeneous modelzoos. For modelzoos of $Arch_2$ in Tables 4 & 5, we maintain the original channels of the datasets. The cross-architecture transfer task is from $Arch_1$ to $Arch_2$. Note also that for the same dataset, *i.e.*, CIFAR10, the cross-architecture evaluation is also from models trained grayscale to RGB images. All modelzoos were generated using the prodcedure outlined in the Appendix of Unterthiner et al. (2020).

Table 3: $Arch_1$ for MNIST, FashionMNIST, CIFAR10 and SVHN.

| Output Size | Layers |
|---|---|
| $1 \times 32 \times 32$ | Input Image |
| $16 \times 30 \times 30$ | Conv2d(in_channels=1 , out_channels=16, kernel_size=3), ReLU |
| $16 \times 28 \times 28$ | Conv2d(in_channels=16, out_channels=16, kernel_size=3), ReLU |
| $16 \times 26 \times 26$ | Conv2d(in_channels=16, out_channels=16, kernel_size=3), ReLU |
| $16 \times 1 \times 1$ | AdaptiveAvgPool2d(output_size=(1, 1)) |
| 16 | Flatten |
| 10 | Linear(in_features=16, out_features=10) |

Table 4: $Arch_2$ for MNIST.

| Output Size | Layers |
|---|---|
| $1 \times 28 \times 28$ | Input Image |
| $8 \times 24 \times 24$ | Conv2d(in_channels=1 , out_channels=8, kernel_size=5) |
| $8 \times 12 \times 12$ | MaxPool2d(kernel_size=2, stride=2), ReLU |
| $6 \times 8 \times 8$ | Conv2d(in_channels=8 , out_channels=6, kernel_size=5) |
| $6 \times 4 \times 4$ | MaxPool2d(kernel_size=2, stride=2), ReLU |
| $4 \times 3 \times 3$ | Conv2d(in_channels=6 , out_channels=4, kernel_size=2), ReLU |
| 36 | Flatten |
| 20 | Linear(in_features=36, out_features=20), ReLU |
| 10 | Linear(in_features=20, out_features=10) |

Table 5: Arch$_2$ for CIFAR10 and SVHN.

| Output Size | Layers |
|---|---|
| $3 \times 28 \times 28$ | Input Image |
| $8 \times 24 \times 24$ | Conv2d(in_channels=3 , out_channels=8, kernel_size=5) |
| $8 \times 12 \times 12$ | MaxPool2d(kernel_size=2, stride=2), ReLU |
| $6 \times 8 \times 8$ | Conv2d(in_channels=8 , out_channels=6, kernel_size=5) |
| $6 \times 4 \times 4$ | MaxPool2d(kernel_size=2, stride=2), ReLU |
| $4 \times 3 \times 3$ | Conv2d(in_channels=6 , out_channels=4, kernel_size=2), ReLU |
| 36 | Flatten |
| 20 | Linear(in_features=36, out_features=20), ReLU |
| 10 | Linear(in_features=20, out_features=10) |

## D   DATASET DETAILS

Dataset splits for modelzoos of Arch$_1$ is given in Table 6. For the cross-architecture task, we generate modelzoos of with 750 neural networks of Arch$_2$ for testing.

Table 6: Dataset splits for modelzoos of Arch$_1$.

| Modelzoo | Train set | Validation set | Test set |
|---|---|---|---|
| MNIST | 11998 | 3000 | 14999 |
| FashionMNIST | 12000 | 3000 | 15000 |
| CIFAR10 | 12000 | 3000 | 15000 |
| SVHN | 11995 | 2999 | 14994 |

## E   HYPERPARAMETERS

We elaborate all the hyperparameters used for all experiments in Table 7.

Table 7: Hyperparameters for all experiments.

| Hyperparameter | Value |
|---|---|
| LR | $1e - 4$ |
| Optimizer | Adam |
| Scheduler | Multistep |
| Batchsize | 64 |
| Epochs | 300 |
| Metric | Binary Cross Entropy |
| NN Encoding Size | 1024 |
| SAB Hidden Size | 512 |
| PMA Seed Size | 1024 |
| # SAB Blocks | 2 |
| chunksize | 32 |
| SAB LayerNorm | False |

## F   IMPLEMENTATION DETAILS

SNE is implemented using Pytorch (Paszke et al., 2019). The SNE model consists of 4 sub-modules:

- **Layer Chunk Encoder:** This consists of two SAB modules where each SAB module is as stack of two SAB layers, followed by a single PMA layer. The layer chunk encoder encodes all the chunks of a given layer independently.

- **Layer Encoder:** This module encodes all the encoded chunks of a layer and consists of two SAB modules and a single PMA layer.

- **Separated Layer Encoder:** This module encodes all the encodings of a layer, for instance the weights and biases, into a single layer encoding vector. It also consists of two SAB modules and a single PMA layer.
- **NN Encoding Layer:** This module takes as input all the layer encodings and compresses them to obtain the neural network encoding which is used for the downstream task of predicting the neural network generalization performance. It also consists of two SAB modules and a single PMA layer.

In addition to the sub-modules above, the layer/level positional encoders are applied to each sub-module when required (see Section 3). The neural network performance predictor, which takes as input the neural network encoding vector from SNE and predicts the performance is detailed in Table 8.

Table 8: Generalization Performance Predictor.

| Output Size | Layers |
|---|---|
| 1000 | Linear(in_features=1024, out_features=1000), ReLU |
| 1000 | Linear(in_features=1000, out_features=1000), ReLU |
| 1 | Linear(in_features=1000, out_features=1), Sigmoid |

