# OpenReview forum: "Set-based Neural Network Encoding"
_ICLR.cc/2024/Conference — Submitted to ICLR 2024_

### Official Review · Reviewer_Yfht · 2023-10-23

**Soundness:** 1 poor
**Presentation:** 3 good
**Contribution:** 2 fair
**Rating:** 3
**Confidence:** 5

**Summary:**

The authors propose an approach to take neural network parameters as input. They treat the parameters as a set and process it with a variant of transfomers. In order to disambiguate weights of different layers with appropriate position encodings based on the layer type and layer number. Chunking of weights is used in order to make the method practical in terms of execution speed.

**Strengths:**

- SNE is a generic technique that is mostly (see second weakness) architecture-agnostic. This is a notable difference from recent work and as such a valuable contribution. This makes the method more widely applicable than prior work.
- Results on the newly proposed datasets appear decent, good experimental evaluation on the proposed datasets.

**Weaknesses:**

- The method is not "minimally equivariant" (see [1] for the definition). In essence, there exist permutations of the weights which *do* change the function, but are considered equivalent by SNE. Consider the flattened weights [1 2 3 4 5 6] with a chunk size of 3. The weights [4 5 6 1 2 3] would be considered exactly the same, since the set of chunks {[1, 2, 3], [4, 5, 6]} are the same. As such, the proposed SNE is not capturing the correct equivariance of the task: it is unable to distinguish between models that are clearly different. This is the primary reason why I'm giving a 1 in terms of soundness. Given the lack of discussion and evaluation of this issue in the paper, I cannot recommend acceptance at this point.
- The authors do not consider the case of architectures with branches and how the position encoding needs to be adapted for them
- There is a lack of comparison against existing methods on established datasets. For example, there are no experiments on implicit neural representations, which DWSNet [2] and NFN [Zhou et al., 2013] are evaluated on extensively. There is no reference to DWSNet in the text.

[1] Neural Functional Transformers. Allan Zhou, Kaien Yang, Yiding Jiang, Kaylee Burns, Winnie Xu, Samuel Sokota, J. Zico Kolter, Chelsea Finn https://arxiv.org/abs/2305.13546

[2] Equivariant Architectures for Learning in Deep Weight Spaces. Aviv Navon, Aviv Shamsian, Idan Achituve, Ethan Fetaya, Gal Chechik, Haggai Maron https://arxiv.org/abs/2301.12780

**Questions:**

Can you state the specific equivariance that SNE has in more detail and the relationship as to how accurately it models the symmetry group of parameters of neural networks?
How does the position encoding work when the model has branches, and as such, there is no total order on the layers?

---

> ### Author Response · Authors · 2023-11-16
> **Response**
>
> We thank the Reviewer for the constructive feedback. We respond to the raised questions below.
>
>
> ***There is a lack of comparison against existing methods on established datasets. For example, there are no experiments on implicit neural representations, which DWSNet and NFN are evaluated on extensively. There is no reference to DWSNet in the text.***
>
>
> Firstly, we reference DWSNet in the text in Section 2(Neural Network Performance Prediction from Weights). With regards to experiments on INRs, we provide the following additional experimental results and benchmark against both DWSNet and the newly introduced NFN models.
>
>
> Experimental setting: We utilize a modelzoo consisting of INRs fit to sine waves on [-$\pi$, $\pi$] and with frequencies sampled from U(0.5, 10). The goal is to predict the frequency of a given INR. All methods encode the INR to a 32 dimensional vector which is then fed into a classifier with 2 linear layers with hidden dimension of 512. All methods are trained for 100 epochs. We report the mean squared error and parameter counts for all models in the table below. We are unable to obtain a converged model for [2] hence we compare with [4] which is the second iteration of the method proposed in [2]. As can be seen,  SNE performs better than the baselines and is parameter efficient compared to [3] and [4]. We note that increasing parameter counts for the MLP and [1] baselines results in overfitting and poor results.
>
>
>
>
>
>
> | Method | Parameter Count | MSE |
> |---|---|---|
> | MLP | 14K | 1.9167$\pm$0.2407 |
> | STATNET[1] | 44K | 0.9373$\pm$0.2761 |
> | NFT[4] | 6M | 0.4010$\pm$0.1085 |
> | DWSNet[3] | 1.5M | 0.2086$\pm$0.0263 |
> | SNE | 358K | $\textbf{0.0977}$ $\pm$0.0024 |
>
>
>
>
>
>
> ***The authors do not consider the case of architectures with branches and how the position encoding needs to be adapted for them***
>
>
> Our discussion ignores architectures with branches, such as ResNets, because we are currently unaware of any standard modelzoo of such architectures on which to benchmark on. However, we provide an outline, in the specific case of residual blocks, of how our method can be adapted to encode such networks.
>
>
> Given that residual blocks are composed of convolutional and linear layers, each of these can be encoded independently as we already do. To account for residual connections, we propose to introduce a special token (just as was done for layer types) when we encode the entire block. While we do not see any impediment to applying our method to such networks, the unavailability of such modelzoos makes it impossible to test this out hence we leave it as future work when such modelzoos become publicly available.

---

> > ### Author Response · Authors · 2023-11-16
> > **Response**
> >
> > ***The method is not "minimally equivariant" (see [1] for the definition). In essence, there exist permutations of the weights which do change the function, but are considered equivalent by SNE. Consider the flattened weights [1 2 3 4 5 6] with a chunk size of 3. The weights [4 5 6 1 2 3] would be considered exactly the same, since the set of chunks {[1, 2, 3], [4, 5, 6]} are the same. As such, the proposed SNE is not capturing the correct equivariance of the task: it is unable to distinguish between models that are clearly different. This is the primary reason why I'm giving a 1 in terms of soundness. Given the lack of discussion and evaluation of this issue in the paper, I cannot recommend acceptance at this point.***
> >
> >
> > With regards to ‘minimal equivariance’, ie. respecting only a subset of permutations in the weight space that are functionally equivalent, there are two approaches to satisfying this property.
> >
> >
> > - Through architecture: This approach is adopted by [2,3,4] where minimal equivariance is baked into the neural network encoder. However, this has a couple of drawbacks:
> >     - It requires a specification of the encoded neural network beforehand. This results in encoders that cannot  be applied to different architectures as we demonstrate in Section 4.2 where at test time, we evaluate our model on architectures not seen during training.
> >     - It requires processing all layers of the network together as opposed to the layer-wise encoding scheme that we adopt. Note that for large networks, [2,3,4] will require correspondingly large memory but the memory requirements of SNE stay constant since we share the same encoder across all layers allowing our model to process arbitrary networks.
> >
> > 	Our goal in designing SNE is to avoid these two drawbacks and hence adopt a regularization based approach to minimal equivariance.
> >
> >
> > - Through Regularization: Recent works[5,6,7,10] have shown that approximate invariance/equivariance can be achieved through regularization during training. This regularization ensures that the models are approximately invariant/equivariant to the desired action group. In the paper, we elected not to discuss this since we found no difference in performance between the regularized SNE model and the unregularized version on the CNN benchmarks[1,2,3,4] for the tasks in Table 1 and 2.
> >
> >
> > However, we present an ablation on the INR task where regularization towards minimal equivariance improves the performance of  SNE. We regularize the encoder by minimizing the distance between functionally equivariant copies of the same network forcing the model to learn the correct minimal equivariance from data[5].
> >
> >
> >
> >
> > | Method | MSE |
> > |---|---|
> > | w/o regularization |  0.1595$\pm$0.0144 |
> > | SNE  | $\textbf{0.0977}$ $\pm$0.0024 |
> >
> >
> >
> >
> > Interestingly, the regularization based approach does indeed perform better than the architecture based approaches. We will add a section on minimal equivariance to explicitly discuss this property.
> >
> >
> > Finally, we draw the Reviewer’s attention to [8,9] where models operating on neural network weights for different problems ignore minimal equivariance across a range of architectures( and especially MLPs) and show state of the art performance. While minimal equivariance is an important theoretical property, in practice, models that violate these properties or even satisfy it approximately are shown to perform similarly or even better than models that strictly satisfy this property.

---

> > > ### Author Response · Authors · 2023-11-16
> > >
> > > **References**
> > >
> > >
> > > [1] Unterthiner, Thomas, et al. "Predicting neural network accuracy from weights." arXiv preprint arXiv:2002.11448 (2020).
> > >
> > >
> > > [2] Navon, A., Shamsian, A., Achituve, I., Fetaya, E., Chechik, G., & Maron, H. (2023). Equivariant architectures for learning in deep weight spaces. arXiv preprint arXiv:2301.12780.
> > >
> > >
> > > [3] Zhou, Allan, et al. "Permutation equivariant neural functionals." arXiv preprint arXiv:2302.14040 (2023).
> > >
> > >
> > > [4] Zhou, Allan, et al. "Neural Functional Transformers." arXiv preprint arXiv:2305.13546 (2023).
> > >
> > >
> > > [5] Cohen-Karlik, Edo, Avichai Ben David, and Amir Globerson. "Regularizing towards permutation invariance in recurrent models." Advances in Neural Information Processing Systems 33 (2020): 18364-18374.
> > >
> > >
> > > [6] Kim, Hyunsu, et al. "Regularizing Towards Soft Equivariance Under Mixed Symmetries." arXiv preprint arXiv:2306.00356 (2023).
> > >
> > >
> > > [7] Otto, Samuel E., et al. "A Unified Framework to Enforce, Discover, and Promote Symmetry in Machine Learning." arXiv preprint arXiv:2311.00212 (2023).
> > >
> > >
> > > [8] De Luigi, Luca, et al. "Deep learning on implicit neural representations of shapes." arXiv preprint arXiv:2302.05438 (2023).
> > >
> > >
> > > [9] Schürholt, Konstantin, et al. "Hyper-representations as generative models: Sampling unseen neural network weights." Advances in Neural Information Processing Systems 35 (2022): 27906-27920.
> > >
> > >
> > > [10] Miyato, Takeru, Masanori Koyama, and Kenji Fukumizu. "Unsupervised learning of equivariant structure from sequences." Advances in Neural Information Processing Systems 35 (2022): 768-781.

---

> > > > ### Author Response · Authors · 2023-11-22
> > > >
> > > > As the Reviewer-Author discussion period closes soon, we would like to kindly request the Reviewer to reconsider based on our response to the
> > > > questions raised. We have made an effort to provide clarifications where necessary and also provide additional experimental results to demonstrate the
> > > > utility of the proposed method. Your feedback is crucial for improving our work and would appreciate your response to the rebuttal. Thank you.

---

### Official Review · Reviewer_8ZLb · 2023-10-24

**Soundness:** 3 good
**Presentation:** 3 good
**Contribution:** 4 excellent
**Rating:** 8
**Confidence:** 3

**Summary:**

This paper focuses on solving the problem of predicting the model accuracy performance given only access to the model parameters. The authors proposed a new method to encode the neural network parameters with set-to-set and set-to-vector functions. The proposed method encodes the model layer types and the layer positional information to retain the model order properties. Experiments show that the proposed SNE not only outperforms existing baselines in terms of the model accuracy prediction correlation, but also has good generalization ability cross different datasets and architecture.

**Strengths:**

- The paper is easy to follow.
- The experiments are convincing and the results are good.

**Weaknesses:**

- The method part is a little bit hard to follow because of so many notations involved.
Please refer to the questions part for more details. Thank you.

**Questions:**

- In Section 3.2, when applying flattening, padding, and chunking operations on a convolutional layer, they are applied on the kernel dimension. I'm wondering why is the case? I have the question because really we have $3\times3$ kernel size for Convs. Do we really need to chunk it anymore?
- Why Equation 2 will convert a vector in size of $c\times1$ to $c\times h$? As mentioned in Section 3.6, $\Phi_{\theta_{1}}$ is SAB(SAB(X)), while SAB(X) is MAB(X,X) in Equation 10. It seems that MAB(X,X) should have the same size as X. How can we get from  $c\times1$ to $c\times h$? You may want to add the dimensions for Equations 10 to 13 for clarification.
- To my understanding, the encoding process from Section 3.3 to 3.5 is like a nested process. Is it possible to provide a pseudocode for it?
- It seems that the "specific dataset $d$" in Section 3.1 is never used. You may want to get rid of it.
- About the experiments, could you have some results on popular CNN architectures like ResNet? Residual links are nowadays important part of the CNN models.

---

> ### Author Response · Authors · 2023-11-16
> **Response**
>
> We thank the Reviewer for taking the time to provide feedback for improving the paper. We respond to the questions raised below.
>
>
> ***In Section 3.2, when applying flattening, padding, and chunking operations on a convolutional layer, they are applied on the kernel dimension. I'm wondering why is the case? I have the question because really we have $3 \times 3$ kernel size for Convs. Do we really need to chunk it anymore?***
>
>
> For convolutional kernels, they are small enough such that chunking is not necessary. We have made the presentation general since the set-to-set and set-to-vector functions are shared across all layer types(linear and convolutions). The chunking operation becomes more important for linear layers to avoid processing large tensors. We will clarify this more in the experiment sections.
>
>
> ***Why Equation 2 will convert a vector in size of $c \times 1$  to $c \times h$? As mentioned in Section 3.6 $\Phi_{\theta_{1}}$ is SAB(SAB(x)), while SAB(X) is MAB(X,X) in Equation 10. It seems that MAB(X,X) should have the same size as X. How can we get from $c \times 1$ to $c \times h$. You may want to add the dimensions of Equation 10 to 13 for clarification.***
>
>
> SAB(X) correctly has the same size of X. However we introduce a projection MLP in the SAB block that projects its inputs to $h$. We will clarify this explicitly in the choice of set-to-set and set-to-vector section and add dimensions to Equations 10 and 13 as suggested.
>
>
> ***To my understanding, the encoding process from Section 3.3 to 3.5 is like a nested process. Is it possible to provide pseudocode for it?***
>
>
> Yes Sections 3.3-5 involve a nested process. We will include pseudocode as suggested and also provide a reference implementation for easy reproducibility and readability.
>
>
> ***About the experiments, could you have some results on popular CNN architectures like ResNet? Residual links are nowadays important part of the CNN models.***
>
>
> While we do not see any impediment to applying our method to networks where residual connections are important, we are currently unaware of any publicly available modelzoos with the appropriate meta-data to benchmark on. An important encoding step in such models will be incorporating the information of residual connections in the encoding pipeline. For this, we propose to introduce a special token(just as was done for layer types) when encoding the entire residual block. We agree with the Reviewer that adopting the proposed method to such architectures is important and we leave it as future work when such modelzoos become publicly available.

---

### Official Review · Reviewer_dyJx · 2023-10-30

**Soundness:** 2 fair
**Presentation:** 2 fair
**Contribution:** 2 fair
**Rating:** 3
**Confidence:** 3

**Summary:**

This paper presents a method of encoding NN architectures by using the trained parameters to predict the validation accuracy of a NN. The proposed set-based encoding improves upon prior work that investigated this problem when evaluated on the same set of benchmarks that include 30k different hyperparameter variations of training small NNs on MNIST-level tasks.

**Strengths:**

Interesting methodology for the set-based encodings. More general than prior approaches.

**Weaknesses:**

Very marginal empirical improvements. Also, I find the motivation for predicting post-training network performance quite lacking. Why is the evaluation on a small validation set insufficient in this case? How does this method compare to things like zero-cost-proxies which try to predict network performance _before_ training?

Additionally, the baseline accuracy for some of these NNs is very low (0.45 on MNIST?)

Minor: SNE is an overloaded acronym in the paper.

**Questions:**

see above

---

> ### Author Response · Authors · 2023-11-16
> **Response**
>
> We thank the Reviewer for the constructive feedback. We respond to the questions raised below.
>
>
> ***Baseline accuracy for some of these NNs is very low (0.45 on MNIST?). Very marginal empirical improvements.***
>
>
> The evaluation metric in Table 2 is **Kendall’s Correlation Coefficient** (as stated in section 4.2. We will explicitly add this to the caption as in Table 1 to avoid further confusion) and not accuracy on the datasets. In table 2, we demonstrate the generality of the proposed method in the cross-architecture task and note that none of the baselines in Table 1 are applicable since the methods are developed for fixed architectures.
>
>
> Across the tasks that we consider, our method performs better than the baselines as shown in tables 1 and 2. But more importantly, the flexibility of the model we propose allows far more capability such as cross-architectural transfer, a task which none of the relevant baselines[1-4] are capable of. Additionally, we provide additional experimental results on implicit  neural representations (INR) below as suggested by Reviewer Yfht.
>
>
> Experimental setting: We utilize a modelzoo consisting of INRs fit to sine waves on [-$\pi$, $\pi$] and with frequencies sampled from U(0.5, 10). The goal is to predict the frequency of a given INR. All methods encode the INR to a 32 dimensional vector which is then fed into a classifier with 2 linear layers with hidden dimension of 512. All methods are trained for 100 epochs. We report the mean squared error and parameter counts for all models in the table below. We are unable to obtain a converged model for [2] hence we compare with [4] which is the second iteration of the method proposed in [2]. As can be seen,  SNE performs better than the baselines and is parameter efficient compared to [3] and [4]. We note that increasing parameter counts for the MLP and [1] baselines results in overfitting and poor results.
>
>
>
>
> | Method | Parameter Count | MSE |
> |---|---|---|
> | MLP | 14K | 1.9167$\pm$0.2407 |
> | STATNET[1] | 44K | 0.9373$\pm$0.2761 |
> | NFT[4] | 6M | 0.4010$\pm$0.1085 |
> | DWSNet[3] | 1.5M | 0.2086$\pm$0.0263 |
> | SNE | 358K | $\textbf{0.0977}$ $\pm$ 0.0024 |
>
>
>
>
> ***Why is the evaluation on a small validation set insufficient in this case? Also, I find the motivation for predicting post-training network performance quite lacking.***
>
>
> While in this work, and those of [1,2,3,4], we have focused on network performance prediction, the methods developed are not limited to this task only. For instance, there are properties of networks[1], such as the type of optimizer, the learning rate, the number of epochs the model was trained for, etc. which cannot be obtained by having access to the training/test/validation set. Such meta-data of pretrained networks may prove necessary in model selection for fine tuning and hence require a means for predicting them.
>
>
>
>
> ***How does this method compare to things like zero-cost-proxies which try to predict network performance before training?***
>
>
> The formulation of the neural network weight encoding problem([1,2,3,4]) assumes access only to pretrained weights making the domain different from zero-cost-proxies used in NAS that predict performance using architectural information.
>
>
>
>
> **References**
>
>
> [1] Unterthiner, Thomas, et al. "Predicting neural network accuracy from weights." arXiv preprint arXiv:2002.11448 (2020).
>
>
> [2] Navon, A., Shamsian, A., Achituve, I., Fetaya, E., Chechik, G., & Maron, H. (2023). Equivariant architectures for learning in deep weight spaces. arXiv preprint arXiv:2301.12780.
>
>
> [3] Zhou, Allan, et al. "Permutation equivariant neural functionals." arXiv preprint arXiv:2302.14040 (2023).
>
>
> [4] Zhou, Allan, et al. "Neural Functional Transformers." arXiv preprint arXiv:2305.13546 (2023).

---

> > ### Author Response · Authors · 2023-11-22
> >
> > As the Reviewer-Author discussion period closes soon, we would like to kindly request the Reviewer to reconsider based on our response to the
> > questions raised. We have made an effort to provide clarifications where necessary and also provide additional experimental results to demonstrate the
> > utility of the proposed method. Your feedback is crucial for improving our work and would appreciate your response to the rebuttal. Thank you.

---

### Official Review · Reviewer_Ab9j · 2023-11-06

**Soundness:** 2 fair
**Presentation:** 2 fair
**Contribution:** 2 fair
**Rating:** 5
**Confidence:** 3

**Summary:**

This paper proposed a set-based neural network encoder for predicting the generalization performance given access only to the parameter values. They also introduce two novel tasks for neural network generalization performance prediction: cross-dataset and cross-architecture.

**Strengths:**

-- Compared with previous methods, the method is applicable to arbitrary architecture benefit from the set-based encoder which also takes hierarchical computational structure of the neural networks into account.

-- To tackle with the large compute memory requirement, this paper proposed a pad-chunk-encode pipeline to encode the neural network layers efficiently.

-- Two new tasks were introduced for evaluating the prediction of the generalization performance.

**Weaknesses:**

-- The motivation of the paper is not clear enough. For a large model, it will lead to large computation overhead, and for a small model you can just simply run the model on the test set. The necessity of the task should be well discussed.

-- The applications of the proposed SNE is still limited. The architectures and datasets are very simple.

-- Lack of ablation study of their methods, such as the effect of their hierarchical computational structure position encoding.

**Questions:**

-- Are the methods applicable for the network with residual connections? In your settings, just given assess to the parameter values, the neural network with or without residual connection will get the same encoding and get the same prediction result. That is not intuitive.

---

> ### Author Response · Authors · 2023-11-16
> **Response**
>
> We thank the Reviewer for taking the time to offer constructive feedback for improving the paper. We respond to the questions raised below.
>
>
> ***The motivation of the paper is not clear enough. For a large model, it will lead to large computation overhead, and for a small model you can just simply run the model on the test set. The necessity of the task should be well discussed.***
>
>
> While research on encoding neural network weights is relatively new [1,2,3,4], these methods have potential application in areas such as learnable optimizers[5,6], direct utilization of implicit neural representations[7], policy evaluation[8] in reinforcement learning, neural network editing[9,10] etc. However, to make progress in all these applications, encoding of network weights is the first step in this pipeline.
>
>
> While in this work, and those of [1,2,3,4], we have focused on network performance prediction, the methods developed are not limited to this task only. For instance, there are properties of networks[1], such as the type of optimizer, the learning rate, the number of epochs the model was trained for, etc. which cannot be obtained by having access to the training/test/validation set. Such meta-data of pretrained networks may prove necessary in model selection for fine tuning and hence require a means for predicting them.
> To demonstrate the generality of the problem, we provide additional applications on implicit neural representations in our response to your second question.
>
>
>
>
> ***The applications of the proposed SNE is still limited. The architectures and datasets are very simple.***
>
>
> SNE makes no assumptions on architectural choices. [1,2,3,4] all require that an architecture is known and fixed a priori. In section 4.2, we introduce the cross-architectural transfer task for neural network encoding to demonstrate the flexibility of SNE even at test time. The most relevant baselines [1,2,3,4] are all not applicable to this task given that they work only for a fixed architecture. Additionally, in Section 4.1, we have introduced the cross-dataset evaluation task, for the first time, for neural weight network encoding methods.
>
>
> Secondly, with regards to architectures and datasets, we evaluate our method on the standard benchmarks for evaluating neural network encoding methods[1,2,3,4]. In section 4.1, we have generated a new modelzoo of pretrained networks to be able to demonstrate cross-architecture encoding. Currently, we are unaware of any large scale modelzoos of architectures such as ResNets or Vision Transformers with the relevant meta-data that can be used for this task. However, we note that the architecture agnostic formulation of SNE ensures that, when such modelzoos become publicly available, the proposed method will still be applicable.
> Finally, we provide additional results on implicit neural network(INR) encoding to demonstrate that our method is applicable beyond the performance prediction task.
>
>
> Experimental setting: We utilize a modelzoo consisting of INRs fit to sine waves on [-$\pi$, $\pi$] and with frequencies sampled from U(0.5, 10). The goal is to predict the frequency of a given INR. All methods encode the INR to a 32 dimensional vector which is then fed into a classifier with 2 linear layers with hidden dimension of 512. All methods are trained for 100 epochs. We report the mean squared error and parameter counts for all models in the table below. We are unable to obtain a converged model for [2] hence we compare with [4] which is the second iteration of the method proposed in [2]. As can be seen,  SNE performs better than the baselines and is parameter efficient compared to [3] and [4]. We note that increasing parameter counts for the MLP and [1] baselines results in overfitting and poor results.
>
>
>
>
> | Method | Parameter Count | MSE |
> |---|---|---|
> | MLP | 14K | 1.9167$\pm$0.2407 |
> | STATNET[1] | 44K | 0.9373$\pm$0.2761 |
> | NFT[4] | 6M | 0.4010$\pm$0.1085 |
> | DWSNet[3] | 1.5M | 0.2086$\pm$0.0263 |
> | SNE | 358K |$\textbf{ 0.0977}$ $\pm$0.0024 |
>
>
>
>
> ***Lack of ablation study of their methods, such as the effect of their hierarchical computational structure position encoding.***
>
>
> We remove the hierarchical compositional structure and the positional encoding mechanism from SNE and test it on the INR task introduced above. From the table below, we can see that these structures are essential to the performance of the method.
>
>
>
>
> | Method | MSE |
> |---|---|
> | w/o positional and Hierarchical Encoding |  7.45167$\pm$0.7985 |
> | SNE |$\textbf{ 0.0977}$ $\pm$0.0024 |

---

> > ### Author Response · Authors · 2023-11-16
> > **Response**
> >
> > ***Are the methods applicable for the network with residual connections? In your settings, just given access to the parameter values, the neural network with or without residual connection will get the same encoding and get the same prediction result. That is not intuitive.***
> >
> >
> > Given that residual blocks are composed of linear and convolutional layers, each of these can be encoded independently as we already do. To account for residual connections, we propose to introduce a special token (just like we do for layer types) for such connections when we encode the entire block.
> >
> >
> > While we do not see any impediment to applying our method to ResNets, we are currently unaware of any publicly available modelzoos with the required meta-data to benchmark our method on. However, it will be important to adopt network encoders to such architectures when such modelzoos become publicly available.
> >
> >
> > Finally, we note that the introduction of such special tokens for residual connections will resolve the issue of outputting the same embedding for the same network with residual connections removed compared to the original residual network.
> >
> >
> > **References**
> >
> >
> > [1] Unterthiner, Thomas, et al. "Predicting neural network accuracy from weights." arXiv preprint arXiv:2002.11448 (2020).
> >
> >
> > [2] Navon, A., Shamsian, A., Achituve, I., Fetaya, E., Chechik, G., & Maron, H. (2023). Equivariant architectures for learning in deep weight spaces. arXiv preprint arXiv:2301.12780.
> >
> >
> > [3] Zhou, Allan, et al. "Permutation equivariant neural functionals." arXiv preprint arXiv:2302.14040 (2023).
> >
> >
> > [4] Zhou, Allan, et al. "Neural Functional Transformers." arXiv preprint arXiv:2305.13546 (2023).
> >
> >
> > [5] Andrychowicz, Marcin, et al. "Learning to learn by gradient descent by gradient descent." Advances in neural information processing systems 29 (2016).
> >
> >
> > [6] Metz, Luke, et al. "Velo: Training versatile learned optimizers by scaling up." arXiv preprint arXiv:2211.09760 (2022).
> >
> >
> > [7] De Luigi, Luca, et al. "Deep learning on implicit neural representations of shapes." arXiv preprint arXiv:2302.05438 (2023).
> >
> >
> > [8] Harb, Jean, et al. "Policy evaluation networks." arXiv preprint arXiv:2002.11833 (2020).
> >
> >
> > [9] Sinitsin, Anton, et al. "Editable neural networks." arXiv preprint arXiv:2004.00345 (2020).
> >
> >
> > [10] Mitchell, Eric, et al. "Fast model editing at scale." arXiv preprint arXiv:2110.11309 (2021).

---

> > > ### Author Response · Authors · 2023-11-22
> > >
> > > As the Reviewer-Author discussion period closes soon, we would like to kindly request the Reviewer to reconsider based on our response to the
> > > questions raised. We have made an effort to provide clarifications where necessary and also provide additional experimental results to demonstrate the
> > > utility of the proposed method. Your feedback is crucial for improving our work and would appreciate your response to the rebuttal. Thank you.

---

### Meta-Review · Area_Chair_6uAW · 2023-12-28

**Metareview:**

This paper proposes a novel method -- named set-based neural network encoding-- for predicting model performance given the neural network weights.  When evaluated on a benchmark set of neural networks trained MNIST-level tasks, the proposed set-based encoding method performs slightly worse (measured in terms of Kendall rank correlation coefficient) than a previously proposed method. However, when evaluated on the introduced novel task of cross-dataset prediction the method performs slightly better. Moreover, the set-based encoding allows for evaluation in a cross-architecture setting.

Two of the four reviews were not convinced about the motivation for predicting model performance from wights. Others criticized the marginal improvements compared to other models and the missing eqivariance that was archived by other methods. Given theses points, I suppose the impact of the paper is not high enough to be presented at ICLR:

**Justification For Why Not Higher Score:**

I am not convinced about the importance of the task and the proposed method misses equivariance and only improves marginally in some settings over previous methods-.

**Justification For Why Not Lower Score:**

N/A

---

### Decision · Program_Chairs · 2024-01-16

Reject